# Using the theory of change (ToC) to co-design a primary mental healthcare system for older people with common mental disorders in Hong Kong

Tianyin Liu[1] ⬤, Jessie Ho Yin Yau[2], Dara Kiu Yi Leung[3], Wai-chi Chan[4], Siu-man Ng[2], Paul Wong[2], Eric Kwok Lun Yiu[2], Samuel Chan[5], Lesley Cai Yin Sze[1], Edwin Wong[6], Wai-wai Kwok[2], Gloria Hoi Yan Wong[7] and Terry Yat Sang Lum[2,8]

[1]Department of Applied Social Sciences, The Hong Kong Polytechnic University, Hong Kong; [2]Department of Social Work and Social Administration, The University of Hong Kong, Hong Kong; [3]Department of Social Work, The Chinese University of Hong Kong, Hong Kong; [4]Department of Psychiatry, The University of Hong Kong, Hong Kong; [5]Department of Educational Psychology, The Chinese University of Hong Kong, Hong Kong; [6]School of Psychology, University of Birmingham, Birmingham, UK; [7]School of Psychology and Clinical Language Sciences, University of Reading, Reading, UK and [8]Sau Po Centre on Ageing, The University of Hong Kong, Hong Kong

## Research Article

**Keywords:**
mental health; collaborative care; community-based initiatives; online Theory of Change (ToC); older adults

**Corresponding author:**
Terry Yat Sang Lum;
Email: tlum@hku.hk

## Abstract

Population ageing and the limited mental healthcare human resources are widening the service gap between older people's mental healthcare needs and the system's capacity in Hong Kong. Scaling up services through integration into the primary care system remains the main strategy to address unmet needs. In this study, we co-developed a primary mental healthcare system for older people with common mental health disorders with 33 stakeholders, including representatives from the government, primary healthcare services, charities, professionals, service users, caregivers and researchers. The study had three phases, including (1) rapid situational analysis (RSA) and survey to synthesise key elements and challenges of the existing service, (2) three rounds of theory of change (ToC) workshops (online) with stakeholders and (3) reach consensus and finalise the ToC map. A shared vision of ***No Wrong Door in Practice*** was established, operationalised as older adults experiencing improved mental health through integrated services from any entry point. The resulting ToC incorporated two interconnected pathways: (1) *medical-social collaboration* to provide integrated and person-centred care, and (2) *community integration* to empower older persons and carers to seek help and navigate the system confidently. Specific interventions, outcomes and outcome indicators were identified in the Hong Kong context for both pathways.

## Impact statements

Mental health is an integral part of health, yet older adults are particularly susceptible to common mental disorders (CMDs). Hong Kong is a "super-aged society" by the United Nations' definition, with a high demand for mental healthcare in older people, but limited human resources and a fragmented healthcare system. Moreover, the pandemic has exacerbated the need for mental healthcare. By utilising the theory of change (ToC) online, we employed a participatory approach to understand the barriers and reimagine an integrated primary mental healthcare system in collaboration with various stakeholders. The findings of this study reveal that there are two pathways to achieve the vision of a long-term mental health policy that has "no wrong door in practice": (1) medical-social collaboration, and (2) community-level collaboration. Some pilot schemes under both pathways have already been implemented in Hong Kong since the ToC. The online ToC provides a structured methodology to identify the present challenges, effectively leverages resources and suggestions from diverse stakeholders and encourages broad political support.





## Background

With population ageing being the most pervasive global demographic trend (World Health Organization, 2021), the mental health of older people is becoming a pressing public health concern, as highlighted by the high prevalence of mental health issues in this demographic. It is estimated that over 20% of adults aged 60 or above have at least one common mental disorder (CMD), including depression, generalised anxiety disorder, phobias, social anxiety disorder and post-traumatic stress disorder (Karel et al., 2012). Moreover, the pandemic has increased the

known risk factors for mental health problems; in particular, it poses more stress and increases loneliness in older people (Banerjee, 2020). Under these circumstances, the increasing mental healthcare needs of older people require a greater demand for mental healthcare, creating increasing pressure on the existing mental healthcare system. Scaling up services through integration into the primary healthcare system could potentially alleviate tensions and prevent overload in the existing mental healthcare system (Petersen et al., 2019).

Different countries employ various strategies to develop their mental healthcare systems, each with its own strengths and limitations. The United Kingdom (UK)'s National Health Service (NHS) was established in 1948 and initially characterised by a centralised and state-driven approach, with significant grassroots advocacy in recent decades (Powell and Williams, 2024). NHS provides universal access and comprehensive mental health coverage, and the latest NHS long-term plan sets out its commitments to expand the services; however, independent and critical reviews of the system have raised concerns over the gap between the number of people with mental health needs and the number receiving treatment (National Audit Office, 2023). The United States of America (USA) has a complex mental healthcare system developed through both top-down (federal government) and bottom-up (advocacy) approaches that provides a diverse range of services (Odilibe et al., 2024). However, access to mental health professionals in the USA is limited compared to other specialists, millions of people live in mental health service-shortage areas, and the high cost is a critical barrier to treatment (Mark et al., 2011). China's present national mental healthcare system was developed using a top-down approach, guided by the Ministry of Health and the service is primarily hospital-based, with a lack of community-based care (Liu et al., 2011). Furthermore, with historical changes and different paces of development in various regions of China, disparities in mental healthcare have become more significant between cities and rural areas and between community and psychiatric hospitals (Liu et al., 2011).

In many low- and middle-income countries (LMICs), there are significant barriers to developing a mental healthcare system, including scarcity of resources, shortage of mental health professionals and uneven distribution of services (Group, 2007; Saxena et al., 2007). To systematically develop and strengthen mental healthcare systems in LMICs, the theory of change (ToC) method is recommended because it draws on a bottom-up participatory approach to co-design complex interventions and systems, and it is an effective planning tool and evaluation framework (Breuer et al., 2015; Hanlon et al., 2016). ToC brings together a range of stakeholders from different sectors and engages them in a series of iterative workshops to discuss and identify the pathway to change towards a common goal, resulting in a ToC map that describes the logic of the causal pathways leading to the outcomes (Taplin et al., 2013). ToC draws on the experience and expertise of participants, allows for rigorous logical thinking, empowers stakeholders to participate at all stages and documents key indicators that enable the systematic evaluation of processes and outcomes (Hernandez and Hodges, 2006). The ToC workshops also facilitate the development of pragmatic intervention by sharing knowledge, articulating assumptions, assessing the interventions' feasibility across settings along the pathway and clarifying key assumptions about the context (Mason and Barnes, 2007).

Despite gaining popularity in strengthening the mental healthcare system in LMICs (Hailemariam et al., 2015; Hanlon et al., 2017; Kafczyk and Hämel, 2021, 2024; Ojagbemi et al., 2023; Murad and Siraj, 2024), ToC has rarely been used in other settings and for older adults with CMDs and in non-LMIC regions. In the present study, we reported using ToC online as a viable strategy for developing a pragmatic primary mental healthcare system for older people with CMDs in Hong Kong. Specific objectives include (1) identifying and prioritising challenges faced by the Hong Kong mental healthcare system collectively; (2) agreeing on the ultimate vision, the long-term, medium-term and short-term goals of the primary mental healthcare system and (3) designing pathways to achieve the goals and address the challenges.

## Methods

### Study setting

Hong Kong has a population of 7.3 million, and 20.1% of them were aged 65 years and over in mid-2022 (Census and Statistics Department, 2022), which qualified as a "super-aged society" by the United Nations' definition (Koohsari et al., 2018). Despite being a highly developed economy (World Bank, 2025), Hong Kong is significantly underpowered in the mental health workforce. For example, there were only 5.55 psychiatrists per 100,000 population in 2022, compared with 16.8 in Organisation for Economic Co-operation and Development (OECD) countries (World Health Organisation, 2019). Given the scarce mental health resources and the increasing demand from the ageing population, it is crucial to develop an integrated primary mental healthcare system for older adults and improve access to services and quality of care. The ToC workshops and interviews were conducted from April 2022 to March 2023, during which Hong Kong was experiencing its largest community outbreak of COVID-19 since the pandemic began. For public health concerns, all co-design workshops and interviews were organised online via Zoom.

### Participants and recruitment strategies

We used purposive sampling to recruit participants from diverse backgrounds, including government officials, directors and key members from non-governmental organisations for older adult services and mental health services, professional specialists in mental health (psychiatrists, clinical psychologists), professional non-specialists (nurses and occupational therapists), professors from tertiary institutions, key members of the charity institution, caregivers and service users. Participants were recruited from the community through the networks of a holistic mental wellness programme aimed at preventing and treating subsyndromal depression in older people in Hong Kong ("JC JoyAge" project) (Liu et al., 2022). Table 1 summarises the composition and number of participants for the three workshops.

### Design and procedures

Co-design has been recognised as a novel methodology increasingly useful for engaging stakeholders to solve complex problems, particularly in the policy context (Blomkamp, 2018). Drawing on a bottom-up co-design methodology, we employed ToC as the overarching approach. This study draws on the ToC framework and uses an iterative process adapted for online workshops and interviews (Table 2). The study has three phases: (1) preparation with a rapid situation analysis (RSA) and survey, (2) three rounds of ToC workshops and follow-ups and (3) sharing the ToC map, reaching consensus and finalising the map. All online ToC workshops were

**Table 1.** Composition of participants in theory-of-change (ToC) workshops

| Sectors | First workshop | Second workshop | Third workshop |
|---|---|---|---|
| | *N* = 33 | *N* = 27 | *N* = 12 |
| Government officials | 1 | 1 | 0 |
| NGOs for older adult/mental health services | | | |
| Directors | 5 | 3 | 2 |
| Key members | 5 | 6 | 0 |
| Professional specialists | | | |
| Psychiatrists | 2 | 1 | 0 |
| Clinical psychologists | 3 | 2 | 1 |
| Professional non-specialists | | | |
| Nurses | 2 | 1 | 0 |
| Occupational therapists | 3 | 2 | 1 |
| Professors from tertiary institutions | 4 | 4 | 2 |
| Charity institution | 2 | 3 | 3 |
| Caregivers and service users | 6 | 4 | 3 |

recorded for research purposes, and participants provided written or verbal consent before participating in and being recorded.

### Preparation phase

Rapid situational analysis (RSA). Situational analysis is the process of gathering information relevant to a local problem that researchers choose to address (Clarke, 2003). RSA refers to a methodology that combines quantitative and qualitative data collection methods, drawing on various data sources (Renfro et al., 2022). In the preparation phase, the JoyAge team conducted an RSA of older adults with CMDs and the mental healthcare system in Hong Kong. The research team searched PubMed, Google Scholar and the Hong Kong government websites prior to January 1, 2022, for published studies and datasets. They also interviewed six seasoned mental health professionals, including a psychiatrist, a family doctor and a clinical psychologist, to gather their experiences with the mental healthcare system. Three research assistants analysed the data and summarised the results.

Online survey. Invitations and consent forms were sent to relevant stakeholders from diverse backgrounds who also attended the first ToC workshop (*N* = 33). Upon their consent to participate in the study, we shared the RSA results with them. We asked them two questions about their *perception of the current mental healthcare system* in Hong Kong and their *vision for an ideal primary mental healthcare system* (see Table 2, "preparation phase," for the survey questions). Participants expressed their opinions by commenting on an online discussion board (Slack) containing the two questions or by emailing their responses directly. Researchers summarised their responses, conducted a thematic analysis (TA) and presented the most common themes in the first ToC workshop for discussion.

### Theory of change workshop

For public safety concerns during this study, the workshops were held online via Zoom. Unlike the usual practice of conducting full-day or longer workshops, the ToC workshops were condensed into 2-hour sessions to accommodate the conflicting schedules of different stakeholders and account for shorter attention spans in a virtual setting.

In the initial workshop, TYSL and GHYW introduced the concept of ToC and then reviewed the RSA results and participants' responses to the two questions posed prior to the workshop. Discussion options (chatroom and open mic) were available to all participants to raise questions, reflect and respond to the information shared. After generating a few common themes regarding challenges, participants were split into five breakout rooms, each focusing on one to two challenges. Participants were allocated to breakout rooms based on their expressed interest and professional expertise to ensure that discussions were both deep and informed. Each room contained a mix of stakeholders (e.g., a clinician, an NGO director and a caregiver) to foster interdisciplinary dialogue, but was themed around a challenge for which certain expertise was most relevant. This mixed-stakeholder composition within themed groups encouraged rich, multi-perspective discussions on specific issues.

In the Zoom breakout rooms, each group had a facilitator to lead the discussion and one to two research assistants to help take notes and construct the group's ToC diagram. One member from each group was nominated to present their ideas and suggestions on that specific area to other stakeholders after the breakout session. The entire workshop, including the breakout session, was conducted in both Cantonese and English and was recorded in video and audio formats. After the initial workshop, a draft of the compiled ToC map was sent to each group, and the principal investigator's (TYSL) video recording was used to explain the map. Communication with all stakeholders took place through email or telephone calls before the commencement of the second workshop to gather feedback on the compiled ToC map.

The same procedures were repeated in the second ToC workshop, which focused on the logic of the pathways between outcomes from different stages. Based on the discussion in the second workshop, an enriched ToC map was generated and sent to participants for feedback before the final ToC Workshop. The third ToC workshop focused on intervention design, and participants with expertise in intervention design participated. Participants were divided into two groups, each focusing on one level of service integration derived from previous workshops. Placement into these groups was based on participants' primary area of expertise (e.g., those with clinical or policy backgrounds focused on the medical-social pathway, while those from community NGOs focused on the community-level pathway). After working separately in Zoom breakout rooms, the groups reconvened to present their respective pathways in the main room. A whole-group discussion then followed to integrate the two pathways, identify synergies, resolve any conflicts and reach a consensus on the overall model.

Facilitation and reflexivity in online workshops. The online format necessitated specific adaptations from traditional face-to-face ToC workshops. The 2-hour sessions were a pragmatic choice to maximise attendance among busy stakeholders and maintain engagement in a virtual setting. To mitigate the challenge of building rapport quickly, facilitators (senior academics and clinicians with extensive experience in mental health and group work) employed several strategies. These included starting with inclusive icebreakers, using the chat function for parallel discussions, and actively inviting quieter participants, particularly service users and caregivers, to share their perspectives first to help balance power dynamics. The pre-existing network of the "JC JoyAge" project,

**Table 2.** Overview of the co-design process

| Phase | Procedures | Participant(s)/data source | Output |
|---|---|---|---|
| Phase 1: Preparation | Rapid situation analysis | Three experienced researchers in mental health, gerontology and social work research (1) searched PubMed, Google Scholar and Hong Kong government websites before January 1, 2022, for published studies and datasets; and (2) interviewed six seasoned mental health professionals | A summary of the current mental healthcare system in Hong Kong and a comparison with other nations |
| | Online Survey: Collect participants' responses to two questions about the mental healthcare system | 29 participants responded to the questions: 1. Based on your personal experience and the slides we shared, what comes to mind when you think about the current state of the primary mental healthcare system for older people in HK? 2. Based on your expertise and perspective, what comes to mind when considering a desirable primary mental healthcare system for older people? | Stakeholders' perception of the current mental healthcare system in Hong Kong and key elements of a desirable mental healthcare system |
| Phase 2: ToC workshop I | ToC workshop conducted via Zoom, involving a general briefing and small group discussions | 33 participants in total, divided into 5 groups for breakout room discussion | Identified the mission and long-, intermediate- and short-term objectives in designing a mental healthcare system for Hong Kong older adults |
| | | 1. Introduced ToC guiding principles to participants 2. Identify the mission and articulate the goals 3. Identify long-, intermediate- and short-term objectives | |
| | Member checking on the ToC map and collecting feedback | A draft of the ToC map was sent out to all participants for revision | Collected feedback from ToC participants for refining the goals of different stages |
| Phase 2: ToC workshop II | ToC workshop conducted via Zoom, involving a general briefing and small group discussions | 27 participants in total, divided into five groups for breakout room discussion on pathways to achieve the objectives | Identified two pathways to achieve the vision |
| | Informal communication via emails and phone calls | Brief participants who did not join the online discussion, get their feedback and incorporate it into the map | Collected feedback from ToC participants for refining the pathways to achieve the vision |
| | Member checking on the ToC map and collecting feedback | A draft of the ToC map was sent out to all participants for revision | |
| Phase 2: ToC workshop III | ToC workshop conducted via Zoom involving general briefing and small group discussions | 12 participants in total, divided into two groups for breakout room discussion | Identified key pilot studies and interventions to achieve the objectives |
| | | 1. Discuss how different parties can collaborate within each pathway and between two pathways 2. Discuss pilot studies/interventions to achieve the objectives | |
| Phase 3: Finalising | Member checking | Cross-check notes, check the logic and resource mapping | Agreement on the final ToC map of the mental healthcare system |
| | | Sending the map to all participants for final approval | |

from which many participants were recruited, also provided a foundational level of trust.

The facilitation team was aware of potential power imbalances between senior professionals, government officials and service users. Techniques to minimise this included validating all forms of knowledge (professional, practical and lived experience) as equally important and ensuring breakout rooms had a mix of stakeholders facilitated by a neutral researcher. The research team's own interests and beliefs, rooted in a commitment to participatory methods and scaling up mental health services, inevitably influenced the process. In line with reflexive TA, we did not view this as a bias to be eliminated, but as a lens that shaped our interpretation. We actively reflected on our positionality during analysis meetings to ensure the final ToC map accurately reflected the stakeholders' collective vision rather than our preconceived ideas.

### Finalising phase

During the finalising phase, all research team members collectively reviewed the notes, logic and resource mapping to validate the data accuracy in the ToC map. Subsequently, the reviewed ToC map was circulated to all participants for feedback and approval, followed by a phone call or email verification process to ensure data validity. This was an iterative process conducted via email and follow-up phone calls due to scheduling constraints. Feedback was systematically compiled, and any disagreements or suggestions for change were reconciled through further discussion with the relevant stakeholders until consensus was reached. Key feedback centred on the plausibility of the collaborations and the feasibility of implementing the proposed interventions, with stakeholders generally agreeing that the ToC represented a desirable and logical, albeit ambitious, roadmap. Ultimately, consensus was achieved on the final ToC map of the mental healthcare system.

### Data collection and data analysis

Data for the RSA of late-life CMD and services in Hong Kong were obtained from multiple sources, including academic search engines, governmental databases and individual interviews with mental health professionals. Online survey data were extracted from the Slack platform or collected through emails. For the ToC workshops,

three types of data were recorded: audio and video recordings of the workshops, observational notes taken by research assistants during the workshops and qualitative feedback from participants after each workshop.

A core research team was formed to analyse the data, comprising four senior researchers in mental health, social work, gerontology and policy, two experienced clinicians (one psychiatrist and one clinical psychologist) and six experienced research assistants in related fields. A framework analysis approach, used in ToC research (Breuer et al., 2014), was employed to analyse the process documentation and ToC workshop transcripts. This method comprises five key stages: familiarisation, identifying a thematic framework, indexing, charting, mapping and interpretation (Ritchie and Spencer, 2002). The thematic framework was identified using reflexive TA, recognising the researcher's subjectivity and valuing it as integral to the analysis process (Braun and Clarke, 2021). Since the research team members all had relevant experience in mental health, gerontology, social work or policy, their coding of the data was nonetheless influenced by their background, and we did not view it as problematic with the reflective TA process, but valued their perspectives when interpreting the data.

After each round of the ToC workshop, the research team familiarised themselves with the data by reading the notes and transcripts, brainstorming initial codes and the codebook. Themes were derived after coding, and the core research team generated themes using reflective TA, reviewed them and mapped them onto the ToC map

components (e.g., short, intermediate and long-term outcomes, outcome indicators). At least two researchers agreed on one theme, and the conflict was resolved through discussion with a senior team member (TYSL). The ToC map was refined through ongoing internal meetings, individual discussions and email correspondence with stakeholders. Microsoft suites (PowerPoint, Word and Excel) and NVivo 12 were used to organise the data and assist with the analysis.

## Results

### Rapid situational analysis results

Results of the RSA covered the following seven areas: (1) the structure of Hong Kong's population, (2) the prevalence of CMD in older people, (3) the impacts of CMD on individuals and society, (4) the mental healthcare workforce, (5) pathways to mental healthcare in the existing system, (6) policies regarding mental healthcare for older people and (7) stakeholders involved from a bottom-up approach. Critically, the analysis of the existing system (point 5) highlighted significant weaknesses and threats related to primary care, including the fragmentation of services, poor integration of private general practitioners into mental healthcare delivery, and a resulting over-reliance on the strained public specialist system. Details of the RSA results are summarised in Supplementary Table 1, and the pathways to care are visualised in Figure 1, which was shared with participants before the first ToC workshop.

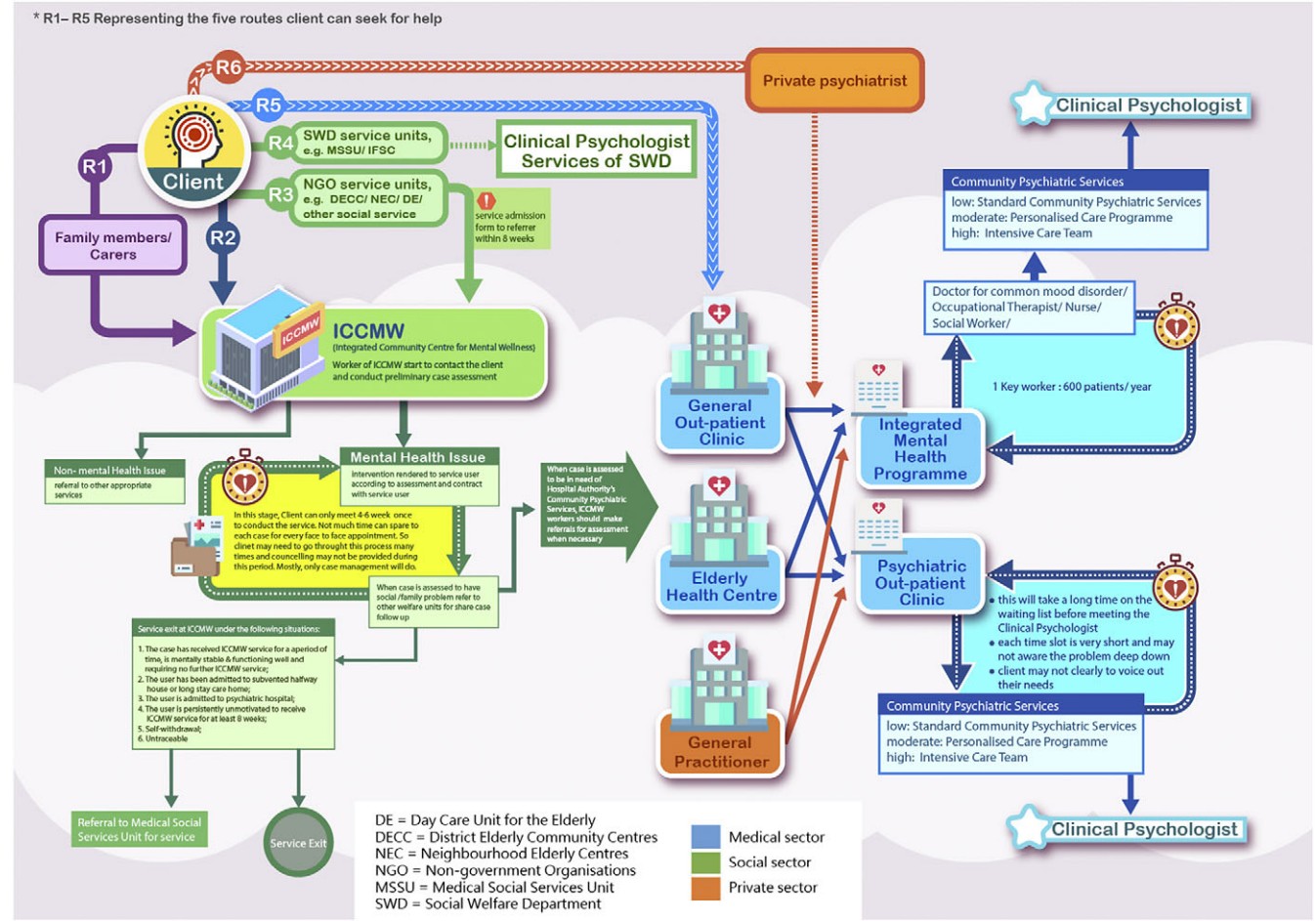

**Figure 1.** Client's journey for older people with common mental disorders.

### Challenges of the current mental healthcare system in Hong Kong

A total of 15 stakeholders responded to the survey question about their perception of the current mental healthcare system in Hong Kong. Having been presented with the RSA findings, their survey responses both validated and expanded upon the initial analysis. Five main challenges emerged from their responses, themes, direct quotes and participants' suggestions are summarised in Supplementary Table 2.

First, all participants mentioned manpower shortage as a significant challenge, echoing the RSA result. The rapid population ageing, long life expectancy and an increasing number of people living with CMD highlighted a rising demand for the mental healthcare system. However, mental healthcare resources in Hong Kong are low compared to other developed regions, highlighting the mental health resource scarcity in the area.

Second, there exists the stigma of mental health in the general public, contributing to low help-seeking. Research suggests that people with mental disorders feel rejected, and their most common coping strategy is remaining secretive (Chung and Wong, 2004; Chung et al., 2019). One participant (a professor in psychiatry with extensive clinical experience) shared his observation from clinical practice that many older adults are resistant to seeking mental healthcare because they may perceive having depression or suicidal thoughts as shameful.

Third, mental health services in Hong Kong are fragmented. In the public sector, various service providers, including the Hospital Authority (HA), the Social Welfare Department (SWD) and non-governmental organisations (NGOs) within the community, each focus on different aspects of services with limited collaboration (Wu, 2021). This leads to the overlapping of services used by frequent service users, who constitute only a small portion of the people in need.

Fourth, the involvement of private primary care services in mental healthcare is inadequate, and people with CMD rely heavily on limited public services, resulting in disproportionately long waiting times. In Hong Kong, general practitioners in the private sector provide a large portion of primary care; however, their motivation to provide mental healthcare to patients with CMD is low. This gap in primary care capacity is a key weakness that directly informed the ToC's focus on a medical-social collaboration pathway. The government heavily subsidises the public mental health specialist services operated by HA. This draws most of the patients, as reflected in the rising number of patients receiving HA mental health specialist services (including inpatients and patients at specialist outpatient clinics [SOPC] and day hospitals) from 186,907 in 2011–2012 to 271,700 in 2020–2021, a 45% increase in less than 10 years (The Government of the Hong Kong Special Administrative Region, 2021). Subsequently, the waiting time for new cases referred to the public psychiatric services has been lengthening over the years, and with HA's priority to treat urgent and semi-urgent cases, the stable cases (i.e., clients who have manageable symptoms, not posing danger to self or others) have to wait an average of for 38 weeks with the longest waiting time of up to 94 weeks for the initial appointment (Hospital Authority, 2023).

Finally, there is no long-term policy on elderly mental healthcare, and this is reflected in the skewed allocation of resources to physical health. However, mental health is a vital and inseparable part of overall health (WHO, 2021), and is interconnected with physical well-being. Poor mental health can increase the risk of physical diseases, and in primary healthcare for older adults, both physical and mental well-being need to be addressed simultaneously.

### ToC map of the primary mental healthcare system for older people with CMD

The results of multiple iterative ToCs are summarised in Figure 2, which describes the final synthesis of all ToC workshops and follow-ups and highlights the two levels of collaborative pathways agreed by participants from different backgrounds. In total, the map outlines 18 key interventions, 6 outcome indicators and 6 assumptions across the two pathways.

#### Impact and key pathways

The ToC workshops enabled us to develop a structured and logical visual map with a collective *vision* agreed upon by participants representing different stakeholders. A common theme repeatedly mentioned in the online survey and the first ToC about the ideal mental healthcare system is one with "No Wrong Door." To put it in one participant's (caregiver) words, "*If I would need mental health help, whichever first step I took, be it GP or local NGO, I wish I will be able to direct to and access the right support that I need.*" This vision converges with the same vision in other countries like the UK (Davies, 2022). One participant (a professor in social work) pointed out that the current mental healthcare system in Hong Kong largely fulfils the "No Wrong Door" vision, but just on paper. Another participant (a client who had used the current system) echoed that statement based on his personal experiences and added that the vision should be "***No Wrong Door in Practice.***" For a more operationalisable impact, participants agreed on a collective vision of older adults in Hong Kong experiencing improved mental health through integrated services from any entry point, supported by a long-term mental health policy in Hong Kong.

With consensus on this vision, long-term (over 10 years), intermediate (5 years) and short-term (2–3 years) outcomes were reverse-engineered and operationalised. The pathways of change consisted of two interconnected and mutually reinforcing levels of collaboration: the medical-social level and the community-level. The medical-social pathway provides integrated services to serve clients, addressing the challenges of manpower shortages, service fragmentation and a lack of private-sector involvement. The community pathway works to increase mental health literacy and direct individuals to appropriate services, addressing the challenge of mental health stigma identified in the RSA and online survey. The two pathways create a synergistic system. Based on these two critical collaboration pathways, participants rearticulated the impact at two levels, and the two pathways to achieve the vision led to two long-term outcomes. Assumptions, interventions and indicators for each pathway were inserted into the preliminary ToC map. A glossary was also attached to define the terms used in the ToC map (Figure 2).

#### Medical-social collaboration pathway

Derived from the main challenges identified in the pre-ToC survey, participants argued that to streamline services for easier access and increase manpower, medical-social collaboration is the most viable solution to mobilise existing resources and integrate them with the existing system. The intended long-term outcome of the medical-social collaboration was that "an integrated, person-centred primary mental healthcare system for older adults is established and functional." During the first ToC workshop, a mental health specialist (a psychiatrist) with extensive experience used the imagery of an "organic funnel" to illustrate the ideal mental healthcare system from his perspective. This system has a broad enough "entrance"

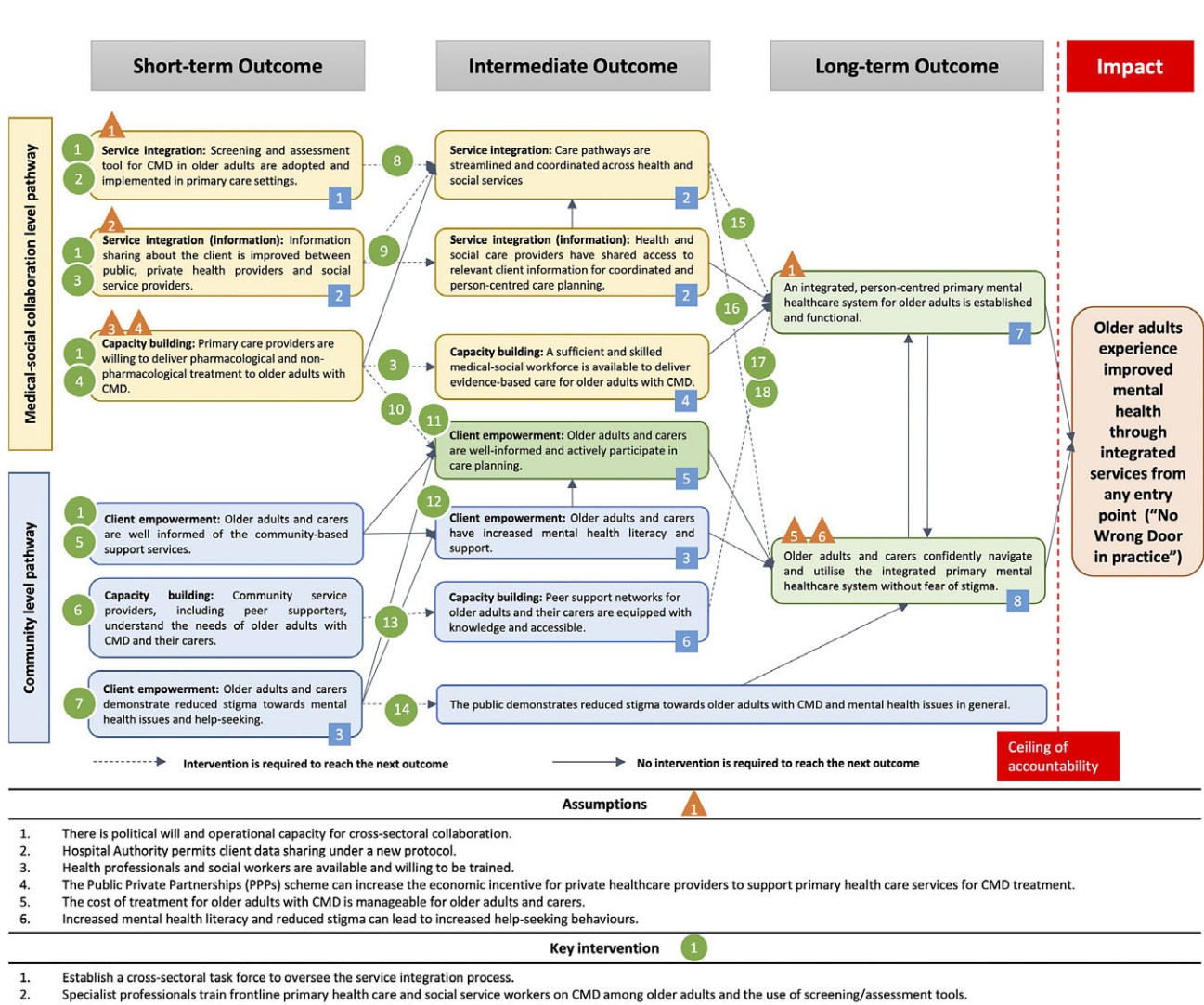

**Figure 2.** Theory of Change (ToC) map on creating a primary mental healthcare system for older people with common mental disorders in Hong Kong.

for screening, then gradually narrows as additional assessments and service matching proceed. Using such imagery facilitated discussions with non-specialist professionals and service users, who shared their views on the dimensions of the "organic funnel" that resonated with them or offered suggestions for improvement. A caregiver shared her experience caring for a relative who had some insight into their health but preferred to see a Traditional Chinese Medicine practitioner. These personal experiences enriched the cultural context of the discussion and highlighted a potential gatekeeper in the medical setting that is rarely mentioned in other societies.

Several interventions were suggested, and a pivotal intervention to achieve short-term outcomes and translate them to the intermediate level was to ensure that a cross-sectoral task force is established and functional. This task force would be responsible for integrating evidence-based non-pharmacological treatments into a unified referral protocol, with its effectiveness assessed through pilot studies. The articulation of these interventions involved iterative rounds and went beyond simply aggregating different opinions. For example, a professor in social work emphasised the importance of research in the design and evaluation of the intervention, and a non-specialist professional raised questions about whether that would interfere with services, which led to a knowledge exchange between different parties. Upon the professionals' and service users' buy-in, they are more open to collaborating with researchers and participating in research.

### Community-level collaboration pathway

The other pathway to impact builds on the success stories and lessons learnt from a local preventive and holistic mental healthcare project ("JC JoyAge"), which implemented a stepped-care model with peer support based on social collaboration and extended to broader community-level service unit collaboration (Liu et al., 2022). The long-term outcome of the community-level collaboration, with synergy at the medical-social level, was that "older adults and carers confidently navigate and utilise the integrated primary mental healthcare system without fear of stigma." This pathway directly addresses individual-level barriers to care by focusing on interventions that improve mental health literacy, challenge self-stigma and promote help-seeking behaviours.

A highlighted dimension of this pathway is the shared intermediate and long-term outcome of reduced stigma of CMD in the general public. Participants recognised the stigma of CMD as a critical barrier to help-seeking, leading to the worsening of undiagnosed symptoms and other dire consequences. In the subgroup discussion, participants noted that stigma was often subtle and more complex to address; therefore, efforts to combat stigma should be long-term, and further research is needed. For now, participants suggested that social service units can conduct more public education as a form of intervention, and more importantly, instead of having professionals or celebrities as spokesmen, invite community champions, that is, service users and persons in recovery, to share their life stories and journeys.

### Discussion

In this study, we described how ToC workshops were adapted online to engage diverse stakeholders and co-design an integrated primary mental healthcare system for older people with CMD in a low-resource setting. The experiences could be used to develop or optimise mental health systems in countries or regions with limited mental health resources.

Compared to the conventional top-down approach to developing a mental healthcare system, ToC has several advantages. First, ToC involves participants from diverse backgrounds, enables knowledge exchange, encourages dialogue about actual needs from users' perspectives and facilitates stakeholder buy-in (Breuer et al., 2014). Besides its participatory nature, ToC provides a valuable framework to guide a deeper level of stakeholder engagement (De Silva et al., 2014). Participants are engaged from the beginning in a formal format for each step, from the vision to all levels of outcomes and, eventually, interventions to achieve the outcomes. There are ample examples in the present study where different stakeholders shared experiences or expressed opinions that were authentic and unique from their perspectives, which were mind-opening to others and stimulated the discussion. However, there were also occasions when professionals dominated the discussion. At those times, facilitators were crucial in bringing awareness to skewed discussions and inviting voices from different perspectives.

Second, ToC has a more flexible format than traditional linear logical frameworks, allowing non-linear and multiple causal pathways to emerge organically from stakeholder discussions (De Silva et al., 2014). The medical-social and community-level pathways identified in the present study were not preset; they emerged as different stakeholders shared their own experiences and opinions about the current mental healthcare system and were reverse-engineered from the shared vision of "no wrong door in practice."

Third, the causal pathways in ToC are contextual and evidence-based, which allows for a detailed understanding of how and whether an intervention might work, which components are most essential and what adjustments to make if needed (De Silva et al., 2014). In addition, we found that moving the workshop online increased the accessibility for participants who were in distant locations or had difficulty travelling. Online conference software and its functions helped streamline communication through screen sharing and real-time messaging. Meetings, including sub-group meetings, could be more easily recorded than the face-to-face format and were well-documented for further review in the follow-up phases. However, the online format may limit the nuanced non-verbal communication and spontaneous relationship-building that characterise in-person workshops. Future research could employ a hybrid model to leverage the accessibility of online platforms while retaining the rich interaction of face-to-face meetings for key consensus-building stages.

Finally, our findings should be discussed in the context of other relevant studies. While much of the ToC literature focuses on LMICs, our study aligns with findings from high-income settings that also grapple with fragmented services. For example, our emphasis on community integration and peer support echoes similar bottom-up initiatives in the UK. The use of ToC for older adults' mental health specifically is nascent. Our process shares similarities with the participatory development of a depression care model for older Nigerians (Ojagbemi et al., 2023), confirming the method's cross-cultural utility for engaging older adults, but our focus on a high-density, super-aged urban environment provides a novel context.

### Policy and programme implications

Engaging diverse stakeholders in the co-design process facilitates collaboration between different sectors and accelerates pilot schemes and policy experimentation. Since the online ToC, the Hong Kong government has announced a pilot project on medical-social collaboration to offer free initial mental health assessments to

members in the pilot districts. The project team is partnering with three District Health Centres (DHC) to provide integrated services to older adults with mental health needs (The Government of the Hong Kong Special Administrative Region, 2024). These developments may promote the institutionalisation of participatory approaches in health policy design in the long term.

Based on the ToC, we offer several recommendations for policy and practice. First, the pilot medical-social collaboration projects should be rigorously evaluated and, if effective, scaled up and formally integrated into the primary healthcare system. Second, dedicated funding should be allocated for community-based public education campaigns, prioritising those led by peers and service users with lived experience to combat stigma effectively. Third, routine mental health screening should be integrated into standard health check-ups for older adults within the primary care setting to facilitate early identification and intervention.

### Limitations

Several limitations need to be considered in interpreting the results of this study. The comprehensive ToC maps contain many details with many smaller processes and critical assumptions to achieve impact. These smaller processes and key assumptions might change due to policy development and, therefore, affect the impact. At a more grassroots level, older adults' knowledge about CMDs, economic constraints and broader socio-cultural factors play a crucial role in determining the choices and decisions they make. Our ToC map did not capture these context-specific factors in a separate pathway, though they are partially addressed through the community integration pathway. Future refinements of the ToC could benefit from developing a dedicated service-user level pathway to more explicitly map interventions targeting these individual factors. Nevertheless, caregivers and service users were included in the ToC workshops to voice their concerns. Another limitation of the ToC approach is that it oversimplifies the complexity of real-world challenges. However, we mitigated this limitations as much as possible by including various professionals in the mental health field. On the other hand, we recommend a follow-up ToC to refine the pathway of change and enhance the elasticity of the map.

### Conclusion

This study shows how online ToC workshops can be conducted to develop a primary mental healthcare system for a specific population with specific needs and contextualised considerations. Using Hong Kong as an example, with limited mental healthcare resources, we illustrate the importance of involving and enabling different stakeholders in co-designing a system that is more likely to be scaled up and facilitates stakeholder buy-in. The whole is greater than the sum of its parts.

**Open peer review.** To view the open peer review materials for this article, please visit http://doi.org/10.1017/gmh.2026.10163.

**Supplementary material.** The supplementary material for this article can be found at http://doi.org/10.1017/gmh.2026.10163.

**Data availability statement.** The data for this study are available from the corresponding author upon reasonable request.

**Acknowledgements.** The authors would like to thank all study participants, workshop coordinators and the research assistants for transcribing the recordings, taking notes, cleaning and organising the data.

**Author contribution.** TYSL and GHYW contributed to the conceptualisation of the project and funding acquisition. TL and TYSL contributed to project administration. WCC, SMN, PW, GHYW and TYSL hosted the online ToC meetings, led the discussion and contributed to data curation. TL, DKYL, EKLY and LCYS assisted in data curation and validation. TL, GHYW and TYSL contributed to the methodology. TL and DKYL led the formal analysis with a research team (JHYY, EKLY, SC and LCYS). TY led the writing of the manuscript. TY, JHYY, EKLY, SC and LCYS contributed to data analysis and original draft. TY and DKYL contributed to data visualisation. GHYW, TYSL, PW, TY and JHYY contributed to the review and editing of the draft.

**Financial support.** This study was funded by the Hong Kong Jockey Club Charities Trust for the University of Hong Kong for the project JC JoyAge: Jockey Club Holistic Support Project for Elderly Mental Wellness (HKU Project Codes AR160026, AR190017). The funder had no role in the design, data collection, data analysis and reporting of this study.

**Competing interests.** The authors do not have any conflicts of interest.

**Ethics statement.** All procedures involving human subjects/patients were approved by the Human Research Ethics Committee (HREC) of the University of Hong Kong. All participants in the ToC workshops and individual interviews provided informed consent to participate.

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
