## [Reviewer Report]

Thank you for the opportunity to review this article. I enjoyed reading this interesting and helpful paper. Please see below more specific comments.

1. Commonly, ToC processes are in face-to-face stakeholder workshops, which allow for a high level of interaction and verbal and non-verbal communication. All your co-design workshops and interviews were organized online, which is a novel approach compared to face-to-face workshops. Given this, the study benefits from more information in the form of reflective evaluation and reflexivity subsection and address issues on: what specific adaptations made to the face-to-face ToC process, the background/experience of ToC facilitators and how this influenced the process, how the online approach have influenced the interactive engagement of the diverse stakeholder groups involved simultaneously in the same ToC process(strategies used, and level of participation of each participant), issues of power imbalance among the diverse stakeholder groups(including the techniques used to minimize power imbalance), how the authors team own interests and beliefs may have influenced the study findings(bias from the researchers involved), and implications for future use of the online ToC co-production compared to the face-to-face approach.

2. Given that your ToC co-production was condensed into two hours, what impact did this have on establishing rapport, building trust, minimizing power imbalance, and active participation of the members of each stakeholder?

3. The way core components in the ToC (both in the narrative and ToC map) are stated/articulated (wording) needs reconsideration. Differentiating outcomes from assumptions/interventions/activities is difficult. The authors did not articulate outcome chains correctly, and the text was heavy in the boxes. Particularly, it would be helpful to reconsider how outcomes were stated at the different levels (short-, intermediate, and long-term). It is difficult to differentiate the long-term outcome and impact as per theory-driven approaches. How are the long-term outcomes at the two major pathways linked (what are their causal links) before implying the “impact”?

4. The findings of your situational analysis did not clearly indicate issues related to primary healthcare (where you are proposing to scale up mental health services): existing opportunities, strengths, weaknesses, challenges, and threats. AND what aspects of these, and how these informed the ToC.

5. Your ToC was focused on two pathways/levels of collaborative pathways as agreed by key stakeholders. However, the findings of your situational analysis (pages 15-17) identified individual level factors (e.g. help seeking behaviour, self-stigma/feelings of rejected), which your raised under the limitation section “… more grass-roots level, older adults’ knowledge about common mental disorders, economic constraints, and broader sociocultural factors play a crucial role in determining the choices and decisions they make” page23, lines 43-50. Don’t you think this suggests the importance of adding a service user levels pathway with necessary interventions to self-stigma, knowledge, attitudes, help-seeking behaviour, preferences

6. What was the basis for placing the participants into the five breakout rooms? Were similar stakeholder groups in the breakout group, or mixed stakeholder groups? How did this affect the engagement?

7. You stated that during the intervention development, the participants were divided into two groups and focused on one level (pathway)... (pp 12, lines 38-41). How did they reach on consensus if each group worked on a different level? What was the basis for placing the groups into the levels?

8. Were all the participants met online to validate the ToC in phase 3, or only through email and phone call? How are differences reconciled? What was their feedback in terms of plausibility, feasibility, and testability of the revised ToC?

9. I think the number of interventions, indicators, and assumptions made needs to be mentioned in the manuscript to bridge and have an idea without seeing the figure legend

10. The author did not say much about the current research progress on using ToC in their study area in the introduction section. There was little information on how ToC stands out from many similar theory-based approaches, and the rationale for choosing ToC for this study. The discussion section also needs to be discussed in line with contextually relevant empirical studies that used ToC for older people, rather than exclusively focusing on the general theoretical benefits of ToC in general contexts.

---

## [Reviewer Report]

Congratulations on a thorough and well written account of your ToC for mental healthcare of older adults in Hong Kong.

Your approach of including the rapid situational analysis gave excellent context and how you applied the ToC approach was methodologically rigorous and well described. You included a broad range of participants and the inclusion of qualitative themes and quotes in your results helped to ground the paper in the responses of participants. The description of the interaction between participants also showed some of the mutual learning taking place in the workshops. This is difficult to achieve in online workshops and indicates to me that the facilitation encouraged participation and engagement between participants.

Although the ToC is presented clearly, I think the content need some revision. Specifically, related to the impact statement. To me, a lot of the ToC was describing what happened as a result of the no Wrong Door Policy. I would suggest framing it around the beneficiaries, something like, “Older adults in HK access high quality MH services across multiple service providers”. It doesn’t sound as good but if you think about the causal pathway you are hypothesizing. I also wonder something similar about the long term outcome related to the “funnel”. In your short term outcome you are already screening. I suggest you carefully look at the temporal order of your outcomes. I would also suggest that any outcome that starts with a verb, “establish”, “improve” is an objective rather than an outcome and should be reworded, for example “reducing the stigma of CMD” which is phrased as an objective to “Stigma towards people with CMD is reduced”.

Some other suggestions:

1) The authors conclude their paper with recommendations on the ToC process but I would have liked to see some recommendations based on the ToC and MH car for older adults in HK.

2) The authors write, “Second, ToC has a more flexible format, which makes explicit the causal pathways that are not pre-defined but allows multiple pathways to emerge organically (De Silva et al., 2014).” More flexible than what?

3) The patient journey diagram is beautifully presented but there are lots of acronyms which make it hard for anyone not familiar with the system to understand what is going on. I appreciate you have a glossary but it is still hard to read. I suggest writing in full where space allows or abbreviating to a single word (psychologist instead of CP).

Overall a great paper which can be benefit from some minor revisions.

---

## [Reviewer Report]

This paper has been strengthened further by incorporating the comments from the reviewers. Some small suggestions as you prepare for publication:

1. In the impact statement, the authors refer to ToC: “By utilising the Theory of Change (ToC) online” and “The online ToC provides a structured” I would edit these to “conducting ToC workshops online” and “the online ToC workshops provides…”respectively as ToC itself is not inherently participatory.

2. In the methods section, the authors sometimes refer to the ToC workshop as 2 hours long when, in fact, there were 3x2 hour sessions. It would be good to consistently refer to this as I noted that Reviewer 1 had misunderstood this point.